# Full scale wind turbine performance assessment: a customised, sensor-augmented aeroelastic modelling approach

Tahir H. Malik[1] and Christian Bak[2]

[1]Vattenfall, Amerigo-Vespucci-Platz 2, 20457, Hamburg, Germany
[2]DTU Wind and Energy Systems, Frederiksborgvej 399, 4000 Roskilde, Denmark
**Correspondence:** Tahir H. Malik (tahir.malik@vattenfall.de)

**Abstract.** Blade erosion of wind turbines causes a significant performance degradation, impairing aerodynamic efficiency and reducing power production. However, traditional SCADA based monitoring systems, which rely on operational data from turbines, lack both effectiveness for early detection and quantification of these losses. This research builds on an established turbine performance integral (TPI) method with a sensor-augmented aeroelastic modelling approach to enhance wind turbine performance assessment, focusing on blade erosion. Applying this approach to a distinct multi-megawatt turbine model, the study integrates multibody aeroelastic simulations with real-world operational data analysis. The study identified readily available sensors that were sensitive to blade surface roughness changes caused by erosion. Operational data analysis of offshore wind turbines validated the initial sensor selection and approach. Refined simulations using further virtual sensors quantified the effect size of these sensor's output under different turbulence levels and blade states, employing Cohen's $d$ - a dimensionless metric measuring the standardised difference between two means. For the investigated turbine, findings indicate that sensors such as blade tip torsion, blade root flap moment, shaft moment and tower moments, especially under lower turbulence intensities, are particularly sensitive to erosion. This confirms the need for turbine-specific, controller-informed sensor selection and emphasises the limitations of generic solutions. This research provides a framework for bridging simulation insights with operational data for turbine specific performance assessment, enabling the improvement of condition monitoring systems (CMS), resilient turbine designs and maintenance strategies tailored to specific operating conditions.

## 1 Introduction

Wind energy has emerged as a cornerstone of the global transition towards sustainable power generation, offering a renewable source that aligns with environmental responsibility and economic feasibility. Central to the operational integrity and efficiency of wind turbines are their blades, whose performance is significantly impacted by the condition of their leading edges. Environmental factors coupled with high tip speeds subject these blades to erosion and surface roughening, which reduces the aerodynamic efficiency and thereby decreases their annual energy production (AEP) (Han et al. (2018); Maniaci et al. (2016); Bak et al. (2020); Bak (2022)). It is well understood that even minor surface imperfections can have profound consequences, adversely affecting performance by altering the blade's aerodynamic profile. This phenomenon necessitates a deeper understanding of how blade erosion impacts wind turbine efficiency, with the aim of developing more resilient blade designs and

maintenance strategies for optimising output and enhancing turbine longevity. Therefore, a comprehensive understanding of the impact of blade erosion on wind turbine efficiency is crucial.

The precise quantification of performance changes caused by blade erosion and subsequent repairs has received considerable attention in wind energy research. Investigations, such as those outlined by Malik and Bak (2024b), have illuminated the complex relationship between blade surface condition, aerodynamics, operational dynamics and turbine's efficiency. This research

builds upon those findings and further explores a refined analytical approach that emphasises the nuances of varying turbine control systems. By integrating multibody aeroelastic simulations for performance data analysis, this study aims to provide a more nuanced understanding. A key aspect of this investigation is the use of turbine-generated supervisory control and data acquisition (SCADA) data for performance monitoring. While the value of SCADA data in this context is well-established (Ding et al. (2022); Yang et al. (2014); Badihi et al. (2022); Gonzalez et al. (2019); Butler et al. (2013)), it has become ev-

ident that existing sensor configurations have limitations. This highlights a pressing need for adaptable monitoring strategies tailored to the specific characteristics of each turbine model and its control system, as emphasised by Malik and Bak (2024b). In contrast to methodologies that generalise sensor pair applications across different original equipment manufacturer (OEM) turbine models, this work emphasises the deliberate selection of a controller-specific sensor pair. For instance, using power as a function of generator speed or power as a function of wind speed indiscriminately across turbines can overlook critical

differences in turbine dynamics and control strategies. This strategy emphasises the importance of finding the most suitable sensor pairings for each turbine and associated controller philosophy.

The primary motivation for the preliminary investigation was to determine whether sensors readily available to wind farm owners and operators via SCADA systems could effectively track individual wind turbine performance and more specifically the reduction in power output due to erosion. The question is whether sensors exist that in the real world that can detect

possible reductions in power output, even amidst the unsteady signals present in SCADA data analysis. This study begins with preliminary multibody aeroelastic simulations, using an OEM-provided proprietary model that matches the operational turbines under investigation. With a focus on rudimentary but widely accessible sensors since often there is a limited sensor array available in SCADA systems (Leahy et al. (2019); Yang et al. (2014)). The initial simulations focus on identifying the correct and effective sensor pairs that exhibit significant sensitivity to blade erosion for the turbine and its controller, setting the

foundation for the development of a turbine-specific turbine performance integral (TPI). This approach recognises that while more advanced sensors may be available to OEMs or may be potentially deployable in future turbine designs, it is imperative to first understand the capabilities of the existing sensor configuration. This prioritisation aims to ensure the findings are relevant and can used to improve current wind turbine performance monitoring system. Guided by these simulation insights, the work then analyses a unique dataset covering sixteen horizontal-axis, three-bladed multi-megawatt turbines with a nominal

power between 3 and 4 MW within the same offshore wind farm with an approximate average wind speed of 9.49 m/s. With the knowledge that can be provided, the corresponding Reynolds number, $Re$, can be determined by the rule of thumb, Bak (2023), where $Re$ is proportional to the radius, $R$, of the rotor and between $75,000 \cdot R$ and $150,000 \cdot R$. Thereby $Re$ is around 7 million. Importantly, some of these turbines were commissioned with leading edge protection (LEP) while others were not, providing a valuable comparison point for erosion effects. Spanning January 2015 to November 2023, this dataset allows for

longitudinal investigation of performance changes due to blade erosion, the staggered application of LEP and blade repairs as well as other events in the turbine's history.

Building upon the author's previous analysis, Malik and Bak (2024b), of wind turbine SCADA data to detect performance impact due to various influences such as erosion, this study extends the analysis to include a distinct turbine model from a different OEM, while continuing to investigate seasonal impacts, long term trends and blade erosion's effects. The turbine performance integral (TPI) methodology introduced in the previous study is employed. This reinforces the validity of the seasonal and trend Decomposition using locally estimated scatterplot smoothing (LOESS) (STL) (Cleveland et al. (1990)) approach for turbine performance assessment but also expands the application scope to include a turbine from an alternative OEM. Importantly, the sensor pairs used in this work are distinct from those in the authors previous publication, specifically aligned with the current turbine model and control system, under investigation. Furthermore, this study eploys the turbines' nacelle-mounted anemometers, despite their inherent measurement uncertainties as highlighted in IEC 61400-12-2:2013 (Commission et al. (2013)) and IEC 61400-12-1 (Commission et al. (2017)) standard - which recommends wind speed measurements at various heights, 2.5 rotor diameters upstream of the turbine. This study avoids the use of separate meteorological masts and demonstrates the potential for monitoring individual turbine performance trajectories using either power as a function of wind speed (measured by the turbine anemometer) or, generator RPM as a function of wind speed metrics.

The 'refined' simulation study, while more aspirational in nature, expands the investigation to a broader spectrum of sensors, including those not currently available to owners but potentially accessible to OEMs, as well as conceptual future sensors. This approach utilises multibody simulations to evaluate a wide range of virtual sensors, identifying those with heightened sensitivity to efficiency changes caused by blade erosion. Simulation scenarios are designed to evaluate turbine responses under various conditions, focusing on wind speeds, turbulence intensities and blade condition. This approach, utilising theoretical models, aims to refine sensor selection methodologies and advance the understanding of wind turbine performance dynamics. Additionally, it aims to provide insights that may inform future research directions in turbine monitoring and maintenance strategies.

This study integrates multibody simulation and SCADA measurement analysis, emphasising the necessity of a turbine-specific, controller-informed approach in monitoring turbine performance changes, as opposed to generalised methodologies. The findings highlight the benefits of strategically selected and deployed sensors, informed by proprietary control philosophies. This research intends to encourage collaboration between academics, turbine manufacturers and operators to implement data-driven strategies for improving the accuracy of turbine performance monitoring.

## 2   Method

### 2.1   Preliminary multibody simulations for sensor pair identification

The study's initial phase employed blade element momentum (BEM) based multi-body aero-servo-elastic tool HAWC2, developed by DTU Wind Denmark (Larsen and Hansen (2007)) to identify sensor pairs potentially sensitive to performance changes caused by blade erosion. The focus of the preliminary investigation is on sensors that are readily available via SCADA

systems. This exploration is predicated on the hypothesises that certain sensor pairs, when analysed under simulated erosion conditions, may provide indications of performance decline. The selection of sensors specifically, pitch, generator RPM and power as functions of wind speed, is informed by the turbine and OEM specific proprietary controller settings. This tailored approach, which explicitly considers controller dynamics, represents a departure from methodologies that do not account for these factors.

This work builds upon the authors' previous findings (Malik and Bak (2024a)) by combining multibody aeroelastic simulations with real-world operational data analysis, thus bridging the gap between simulation-based insights and empirical validation. The previous study focused solely on the simulated environment, investigating the combined effects of leading edge erosion and turbulence intensity ($TI$), as well as exploring time-interval averaging as a data processing technique. To assess the feasibility of observing the power degradation in real-world measurements, that study compared the performance of turbines with clean blades to those with simulated surface roughening.

This study uses the same certified OEM-provided certified multibody model of an operational turbine's controller in the full aero-servo-elastic simulation loop ensuring the accurate capture of the response to degraded blades, including pitch adjustments utilising aerodynamic reserves. Furthermore, the previous study advocated for using higher resolution data in analysis to improve the detection of subtle performance changes, a recommendation that this current study implements by utilising 1-second sampled, rather than 10-minuted averaged data. For a more detailed elaboration on the employed methodology and insights offered, readers may refer to the aforementioned paper.

Furthermore, in this work the effectiveness of the identified sensor pairs for investigated turbine is compared to those found effective in previous research Malik and Bak (2024b), where a distinct wind turbine from a different OEM was studied and for which the relationship of generator speed as a function of power formed the basis for monitoring performance variation over time using turbine performance integral (TPI). This cross-turbine sensor comparison reinforces the importance of tailoring sensor selection to specific turbine models and control systems. Furthermore, the validation of the TPI method for the turbine under investigation, demonstrates the methods applicability across diverse wind turbine designs. These elements of the study have the potential to improve the sensitivity and accuracy of performance monitoring across varied wind turbine configurations.

### 2.1.1 Modelling leading edge erosion

To model blade leading edge erosion, a surface roughness based on wind tunnel tests is used from Krog Kruse et al. (2021). These tests utilised P400 (fine) and P40 (coarse) grit sandpaper to simulate different erosion levels on an alternate aerofoil and provided the empirical basis for deriving factors for the blade modifications. To simulate early-stage degradation the outer 15% of the blade model's original aerofoil polars are altered by applying a factor of $0.9$ to the clean aerofoil polar and scaling the drag polar by factors of $1.5$ (P400) and $2.0$ (P40) (see Malik and Bak (2024a) for details) to reflect observed erosion after approximately two years of operation. It is important to note that, relying on relative changes this study employs a simplified approach and the simulated roughness may differ from the actual turbine's conditions. Therefore, while these simulations reflect deteriorating changes in blade conditions, they do not necessarily represent the precise changes that occur in real-world scenarios.

### 2.1.2 Simulation settings and test cases

To analyse the impact of turbulence intensity and blade erosion on wind turbine performance, simulations were conducted using an OEM-provided multibody model representing the operational offshore wind turbine also investigated as part of this work. Simulations were performed for clean and two blade leading edge erosion states across a range of turbulence intensities, with further model parameters and conditions provided in Malik and Bak (2024a). In contrast to the previous work, where simulations were run at 1 m/s increments, the current study employs a higher fidelity approach. To focus on the turbine's power ramp-up phase (where erosion effects are most likely to manifest) and to ensure that the binning and averaging process of the data did not obscure subtle dynamics, individual cases were run in 0.1 m/s increments between 6.5 and 14 m/s. This increment achieves a balance between fine-scale accuracy and computational efficiency. Following the International Electrotechnical Commission (IEC) (2019) 61400-1 standard, six individual simulation runs, or seeds, were used per configuration to ensure statistical robustness.

Turbulence intensity ($TI$) was varied across a spectrum (0%, 3%, 6%, 9% and 12%), with 6% approximating filtered average offshore conditions. Simulations were executed for 900 seconds, with data from the last 600 seconds analysed to ensure steady-state conditions. Time steps were set at 0.01 seconds. Wind shear followed a power-law profile with an alpha value of 0.14 and air density was fixed at 1.225 kg/m$^3$ (representative of sea-level conditions at 15°C). The default Mann turbulence model parameter $\alpha\epsilon^{2/3}$ of 1 was used (Mann (1994)). For detailed explanations, please refer to the HAWC2 manual (Larsen and Hansen (2007)) and IEC61400-1 ed. 3 International Electrotechnical Commission (IEC) (2019).

With a focus of the preliminary investigation on sensors that are readily available via SCADA systems, simulations utilising the multibody aeroelastic model, facilitated the identification of sensor pairs that exhibit significant sensitivity to blade erosion, setting the foundation for the development of a turbine specific turbine performance integral (TPI). Due to confidentiality agreements, a generalised description of the turbine is provided and results are presented in relative terms.

### 2.2 Wind turbine operational SCADA data analysis

Building upon the sensor pairs identified through multibody simulations, this section conducts an analysis of SCADA data from operational turbines. By focusing on the power as a function of wind speed and generator RPM as a function of wind speed sensor pairs, this investigation aims to validate the simulation-derived hypotheses within a real-world setting, assessing their feasibility and effectiveness in detecting blade erosion. This analysis both tests the hypotheses generated from the simulations and provides a practical framework for evaluating the sensor pairs' effectiveness in performance monitoring.

Sixteen front-row, offshore multi-megawatt turbines within the same wind farm were selected for their direct exposure to dominant wind conditions. Due to confidentiality agreements, the specific site or turbine type shall not be disclosed. The wind farm provides a unique SCADA dataset spanning January 2015 to November 2023. This dataset offers a valuable experimental timeline, with some turbines installed with a specific LEP (Type A), while others remained unprotected. As expected, unprotected blades exhibited significantly greater erosion, already within the first two years of operation. Starting in 2019, remedial actions were taken with the repair of unprotected blades and the application of a different shell type LEP system (Type B).

This application was phased, with some turbines receiving partial LEP coverage (approximately 7-8% of the blade span) and others receiving complete coverage (15%). Notably, LEP application could take between a week and, in exceptional cases, up to a month, due to logistical arrangements in an offshore environment. In 2021, the remaining turbines received full LEP coverage. Additionally, minor LEP repairs (approximately 0.5 - 1.5 m) were performed in 2020 and 2021; however, these lesser interventions are not expected to produce an impact measurable in turbine performance. This dataset, with its distinct phases of LEP application and repair, provides an opportunity to investigate the longitudinal effects of blade erosion and the impact of the application of LEP, or change in the aerodynamic profile, on wind turbine performance. Data regarding LEP applications and repairs were obtained directly from technician reports.

From the restricted set of sensors accessible through the SCADA system, the following parameters pertinent to the investigation were gathered:

- Nacelle wind speed $\nu$ (m/s)

- Nacelle direction (°)

- Ambient temperature $T$ (°C)

- Blade pitch angle $\beta$(°)

- Generator speed $\Omega$ (RPM)

- Power production $P$ (kW)

- Power setpoint demand $P$ (kW)

- Turbine operational state (e.g. waiting for wind, curtailed, cable unwind, etc.)

To heighten the accuracy of detecting subtle performance changes (Badihi et al. (2022); Malik and Bak (2024a)), this study utilised a dataset comprising SCADA data sampled at one-second intervals (rather than 10-minute averaged data) which were pre-computed from the wind turbine's data archive, where a sensor's signal is only updated when a change is recorded. Missing values were handled using the 'previous value' method to reduce computational demands. The dataset was filtered and processed according to International Electrotechnical Commission (IEC) 61400-12-1 guidelines Commission et al. (2017), but not corrected for temporal density variations. Nacelle direction served as a proxy for wind direction, despite its influence by the turbine's control algorithm hysteresis and rotor wake.

### 2.2.1 Wind turbine control and turbine performance integral

An understanding of the investigated turbine's characteristics reveals that the turbine employed in this study contrasts with previous work, Malik and Bak (2024b), where the TPI method was first introduced, such that the rotor control does not primarily rely on its wind speed anemometer as a control input during its power generation mode. Once generating power, the

turbine controller relies on operational trajectories following a speed-power and a pitch-power curve rather than using direct information regarding the wind speed. Examples of such control include work by Hansen and Henriksen (2013).

For the investigated turbine, the turbine performance integral (TPI) is defined as the area under the power curve between wind speeds of 6 and 10.5 m/s. This integral, with units of Power.Wind Speed (kW.m/s) is used to extract the seasonal variations using STL technique that serve an indicators of the turbine's performance trajectory. Alternatively, the generator RPM as a function of wind speed area metric (between 5.5 and 8.5 m/s) may be employed. It is important to ensure that the selected wind speed limits create a monotonic relationship and that the turbine operates outside of full load conditions. This is because the effects of erosion are primarily visible in partial load conditions. The pitch angle versus wind speed relationship only becomes monotonic between 10.5 and 11.5 m/s, making it less suitable.

A weekly updating ring buffer with a fixed value is employed, adjustment of which affect TPI outcomes. The structure and data flow of the ring buffer system can be visualised in Figure 1. This block diagram illustrates how sensor data (in this case, power and wind speed output) is input to the system, stored in a ring buffer and processed through bin-wise trapezoidal integration to compute the TPI. Additionally, the diagram shows the data carryover mechanism, where previous week's data is used to fill gaps when insufficient new data is available.

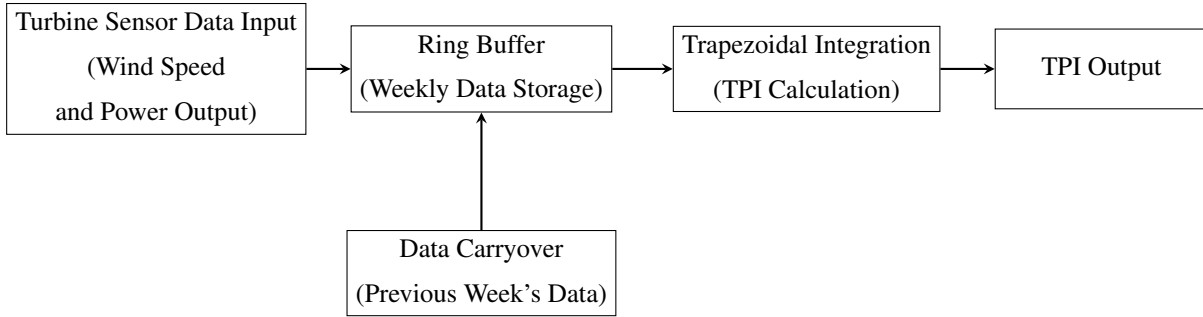

**Figure 1.** Block diagram of the ring buffer system for wind turbine performance monitoring

The ring buffer's mathematical model is based on modular arithmetic, which facilitates its circular structure. Let $B$ represent the *buffer size*, $i_{\text{current}}$ the current index for data entry, and $t_n$ the $n$-th data point from the sensors. The position for the next data point is determined by:

$$i_{\text{next}} = (i_{\text{current}} + 1) \mod B \tag{1}$$

This equation ensures that when the buffer reaches its capacity, it wraps around and starts overwriting the oldest data. The *buffer size* affects how quickly changes are detected. A large buffer may smooth out short-term variations, while a smaller buffer is more responsive to immediate fluctuations.

Once the data is stored in the buffer, the TPI is calculated using trapezoidal integration. TPI quantifies turbine efficiency by representing the area between the power ($P$) as a function of wind speed ($v$) and the wind speed axis over a specified range. Mathematically, the TPI is defined as:

$$\text{TPI} = \int_{v_1}^{v_2} P(v) , dv \tag{2}$$

This integral calculates the area under the power curve between the power levels $v_1$ and $v_2$, which correspond to 6 and 10.5

215   m/s, respectively, providing a measure of turbine performance within this operational range.

### 2.2.2   Seasonal trend decomposition and data visualisation

An analysis of wind turbine SCADA data is used to assess the influence of seasonal effects and blade erosion on performance. This study utilises the approach employed in Malik and Bak (2024b), where the turbine performance integral was first introduced. The TPI signal, is used to extract the seasonal variations using using the seasonal and trend decomposition using LOESS

(STL) method, Cleveland et al. (1990). The STL technique decomposes a time series into three components: seasonal, trend and residual. This decomposition is mathematically represented as follows:

$$Y_t = T_t + S_t + R_t \tag{3}$$

where $Y_t$ denotes the observed data at time $t$, $T_t$ is the underlying performance trend component, $S_t$ is the cyclical seasonal component related to annual variations of atmospheric conditions and $R_t$ is the residual component that is composed of

unattributed transient factors.

      This work focuses on the direct impact of LEP applications and repairs on long-term performance trends. Rather than attempting to isolate the various factors influencing performance, as done in the previous study, this work overlays data regarding LEP applications and repairs onto the long-term performance trajectory. This approach acknowledges the limitations of this approach in providing a comprehensive picture but attempts to offer insights into the direct effects of these interventions. A

multi-panel visualisation with a shared time axis is employed to analyse wind turbine performance data decomposed using STL, which was performed using MATLAB's "trenddecomp" function (The MathWorks, Inc. (2023)). This approach allows for the simultaneous examination of long term trend, seasonal and remainder components, highlighting their interactions over time. The shared temporal axis serves as a reference point to compare the evolution of each component, aiding the identification of changes and potential anomalies within the data.

While previous work, Malik and Bak (2024b), emphasised the meticulous collection of operations and maintenance (O&M) data, including detailed accounts of events that included blade erosion and repair related interventions, the current investigation adopts a more focused approach. This decision does not diminish the significance of O&M activities on turbine performance. Instead, it aligns the scope with the specific objective of validating and applying the TPI method. This approach provides an

illustration of the method's capabilities, within the context of a distinct OEM model and control system, rather than constituting a comprehensive analysis of O&M's influence on turbine performance.

## 2.3 Refined multibody simulations for detailed sensor evaluation

Building upon the empirical validation of initial findings, this research advances to a series of multibody simulations designed to gain a deeper understanding of various sensor's sensitivity to blade erosion under varied turbulence intensity conditions. Details of the simulation methodology may be found in the earlier Section 2.1, where the preliminary investigation is described.

The primary objective of this exercise is to evaluate a diverse array of sensors chosen based on their potential to detect changes in blade aerodynamic performance due to erosion. While a wider selection of sensors was simulated, including lift and drag coefficients at various blade positions, the displayed sensors were down-selected based on the following criteria:

– Relevance to blade aerodynamic performance: Sensors that directly or indirectly measure parameters influenced by changes in blade surface conditions, such as blade loads, power output and moments, are prioritised.

– Availability in existing SCADA or CMS systems: Sensors that are commonly available or can be readily integrated into current monitoring systems are preferred to facilitate practical implementation in real-world scenarios.

– Sensitivity to erosion-induced changes: Sensors that exhibit a clear and measurable response to varying levels of blade erosion are selected to ensure reliable detection.

– Signal-to-noise ratio: Sensors with high signal-to-noise ratios are chosen to minimise the influence of external factors and measurement uncertainties.

While the findings for these sensors may be specific to the studied turbine, the process serves as an example of a procedure that may be followed for other turbines. This evaluation begins with selecting a broader spectrum of virtual sensors within the simulation environment to identify the most reliable indicators of erosion-related performance changes. These sensors include, but are not limited to, blade root bending moments, blade tip deflections, tower top and bottom loads and drivetrain torque. The selection criteria prioritise sensors or data channels that are readily deployable and practical in real-world scenarios and have the potential to improve existing monitoring and performance analysis capabilities.

Next, a series of multibody simulations are conducted, modelling the turbine under various operating conditions. The selected sensors are subjected to a series of simulations under various blade erosion states (clean, P400, and P40) and turbulence intensity conditions (0%, 3%, 6%, 9%, and 12%). The generated sensors response is then processed and analysed using Cohen's $d$ (described in detail in later Section 2.3.1) to quantify the effect size of blade erosion on each sensor's output. Sensors exhibiting high sensitivity are identified as potential candidates for erosion detection and performance monitoring. The insights gained from the simulation results are then discussed in terms their relevance and practical application.

The methodology explores theoretical simulation but stops short of empirical validation, that would ensure that the findings are anchored in both theoretical rigour and operational relevance, due to lack of existence or access to the broader sensor suite in the real world. This exercise, however, exposes the potential of such sensors in revealing critical aspects of turbine

performance and advocates for their inclusion in future turbine designs, which is a key motivation of this study. Despite this, the results are discussed for their practical applicability. This simulation-based methodology aims to complement traditional SCADA data analyses, providing insights that might be difficult to glean from operational turbines alone, while simultaneously highlighting the need for development in sensor deployment in wind turbines to improve performance monitoring and maintenance strategies.

### 2.3.1 Framework for sensor output comparison - Cohen's $d$ calculation

This study quantifies the impact of erosion through differences in sensor output, providing detailed visualisations of both clean and eroded blade states. The primary objective is to gain a deeper understanding of turbine performance dynamics and to enable the development of proactive monitoring strategies for early detection of erosion or performance deviations.

To compare multiple sensor outputs under different blade conditions, a robust statistical metric is needed. Cohen's $d$ (Cohen (1992)) was chosen due to its ability to quantify effect size. It provides a standardised measure of the difference between two means that is independent of the units of measurement. This allows for meaningful comparisons across diverse sensor outputs (e.g., blade root bending moment or tower moment as functions of wind speed).

Crucially, Cohen's $d$ provides a normalised measure of effect size. This is essential for understanding the magnitude of erosion's impact and identifying sensors that are most sensitive to changes in blade aerodynamic surface properties. Importantly, using a percentage change for this comparison would disproportionately emphasise changes in values close to zero, whereas Cohen's $d$ avoids this potential bias.

The Cohen's d was applied in an analysis of full scale measurements, Malik and Bak (2024b) and serves as the link between the simulations and future full scale measurements. Using this method shall indicate whether certain signals can be detected better than others.

To quantify the difference between "clean" and "rough" (P40) blade conditions for each sensor and wind speed bin, Cohen's $d$ was calculated:

$$d = \frac{\overline{x}_{\text{rough}} - \overline{x}_{\text{clean}}}{s_p} \tag{4}$$

where $d$ is Cohen's $d$ (a dimensionless measure of effect size), $\overline{x}_{rough}$ is the mean of the sensor data in the "rough" blade condition, $\overline{x}_{clean}$ is the mean of the sensor data in the "clean" blade condition, $s_p$ is the pooled standard deviation, calculated as:

$$s_p = \sqrt{\frac{(n_{\text{rough}} - 1)s_{\text{rough}}^2 + (n_{\text{clean}} - 1)s_{\text{clean}}^2}{n_{\text{rough}} + n_{\text{clean}} - 2}} \tag{5}$$

where $n_{rough}$ is the number of samples in the "rough" condition, $n_{clean}$ is the number of samples in the "clean" condition, $s_{rough}$ is the standard deviation of the sensor data in the "rough" condition, $s_{clean}$ is the standard deviation of the sensor data in the "clean" condition.

The magnitude of Cohen's $d$ aids in interpreting the practical significance of the differences observed between clean and rough blade conditions. Values around 0.2 indicate a small effect size, 0.5 a medium effect and 0.8 or greater suggest a large effect. However, these values should be interpreted as a guide that should be informed by the context of the relevant sensor in context of this analysis - Cohen (1992). This allows for identifying the most erosion-sensitive sensors and assessing the impact's magnitude.

Furthermore, this metric is particularly well-suited for this work, as it incorporates pooled standard deviation. This accounts for potential variability in the number of data points across simulations and sensors, ensuring robust comparisons.

## 3 Results and discussion

### 3.1 Preliminary multibody simulations for sensor pair identification

The comparative analysis revealed substantial behavioural differences between sensor pairs, attributable to the varying turbine control systems. For the turbine investigated in this study, illustrated in Figures 2 and 3, the relationships between blade pitch angle and generator speed as functions of normalised power, did not exhibit any noticeable changes due to alterations in blade roughness (error bars represent one standard deviation). This finding contrasts sharply with the sensor pair dynamics of the turbine evaluated in Malik and Bak (2024b), where this specific sensor pair formed the basis of the TPI signal.

However, Figures 4 and 5 demonstrate that erosion at the leading edge significantly affects turbine performance. In the former, an eroded blade necessitates more aggressive pitching to sustain power generation, while in the latter, an eroded blade manifests in lower RPMs for any given wind speed. This suggests a shift in operational setpoints, given that the turbine's control algorithm does not incorporate wind speed measurements from its anemometer during production.

These results highlight the necessity for a turbine-specific approach in selecting sensor pairs to effectively assess turbine performance. The inefficacy of a generic, one-size-fits-all strategy is inadequate for addressing the intricacies of diverse turbine control philosophies. Thus, it is imperative to develop tailored sensor pair selection methods to ensure the fidelity of performance integrity evaluations.

Furthermore, shown in Figure 6 is the normalised power curve for three blade profiles. These simulations are executed at 6% $TI$, which approximates the mean annual turbulence intensity where the real offshore turbines analysed later in this study are located. The simulation results clearly demonstrate that the roughening of the blade leading edge has a detrimental impact on the turbine performance. The area under this normalised power curve, specifically between wind speeds of 6 and 10.5 m/s, shall form the foundation of the turbine performance integral (TPI) signal. In this manner the TPI signal encapsulates the variation in power output due to blade surface conditions. It offers a quantifiable metric to assess the degree of erosion's impact on turbine efficiency.

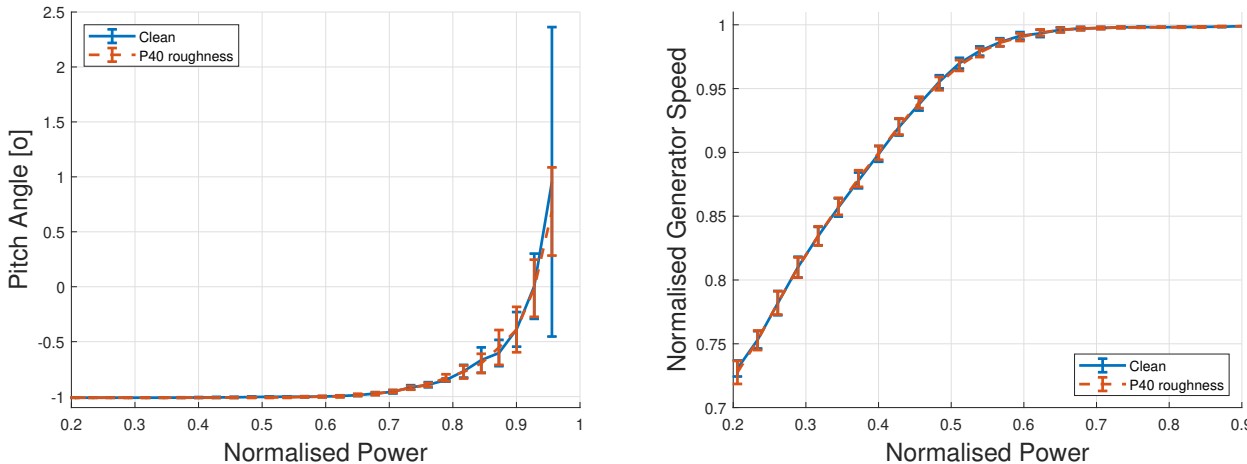

**Figure 2.** Blade pitch angle as a function of normalised power for clean and rough blade profiles, with a fixed turbulence intensity of 6% - Simulated.

**Figure 3.** Normalised generator speed as a function of normalised power for clean and rough blade profiles, with a fixed turbulence intensity of 6% - Simulated.

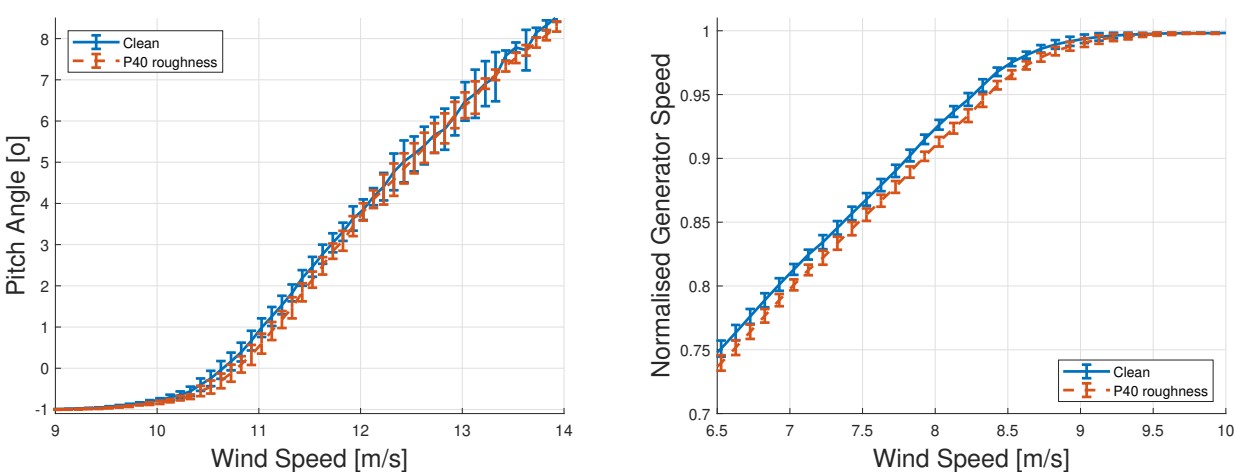

**Figure 4.** Blade pitch angle as a function of wind speed for clean and rough blade profiles, with a fixed turbulence intensity of 6% - Simulated.

**Figure 5.** Normalised generator speed as a function of wind speed for clean and rough blade profiles, with a fixed turbulence intensity of 6% - Simulated.

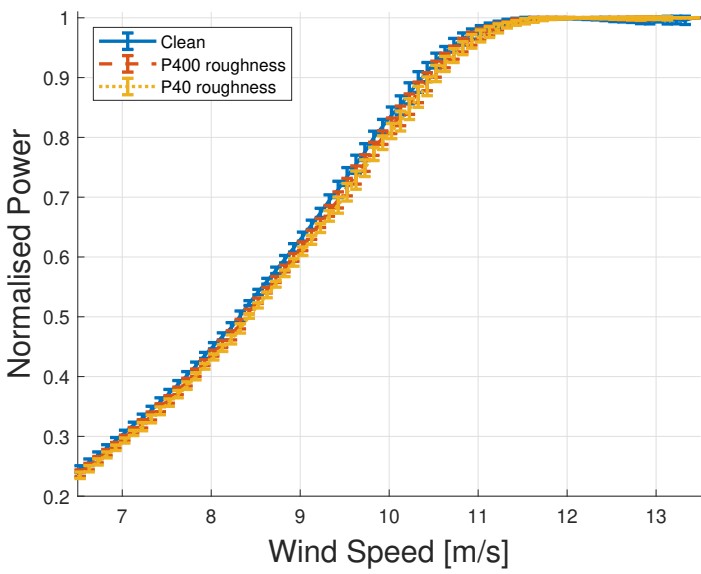

**Figure 6.** Normalised power as a function of wind speed for various blade profiles, with a fixed turbulence intensity of 6% - Simulated.

### 3.2 Wind turbine operational SCADA data analysis

Building upon the foundation of the authors previous work Malik and Bak (2024b), which embarked on a comprehensive effort to correlate turbine performance with operations and maintenance (O&M) events, this study adopts a more focused approach. Recognising the considerable resource investment required to compile comprehensive O&M datasets, particularly those pertaining to blade erosion and repair-related interventions, this investigation focuses on demonstrating the application of the TPI method. This deliberate focus not only validates the decomposition technique for assessing turbine performance but also broadens the framework to incorporate a turbine from an different OEM. Thus, it serves to bridge the findings of previous work Malik and Bak (2024b) with the focused investigations of the current paper.

Presented in Figure 7 is the empirically measured power curve for the turbine in question, with the variability indicated by the standard deviation bars. This dataset spans approximately nine years. For this graphical representation 10-minute averaged data were utilised, whereas all other measurement analysis utilises non-time-averaged 1-second sampled data. The data were filtered and processed in adherence to the standards prescribed in the IEC 61400-12-1 Commission et al. (2017). This 10-minute averaging allows for a direct visual comparison with the simulated power curve shown earlier in Figure 6. Variation between the two curves profiles may be attributed to an array of influences, including the fidelity of data filtering, temporal changes in turbine performance, fluctuating atmospheric conditions and the impact of O&M interventions.

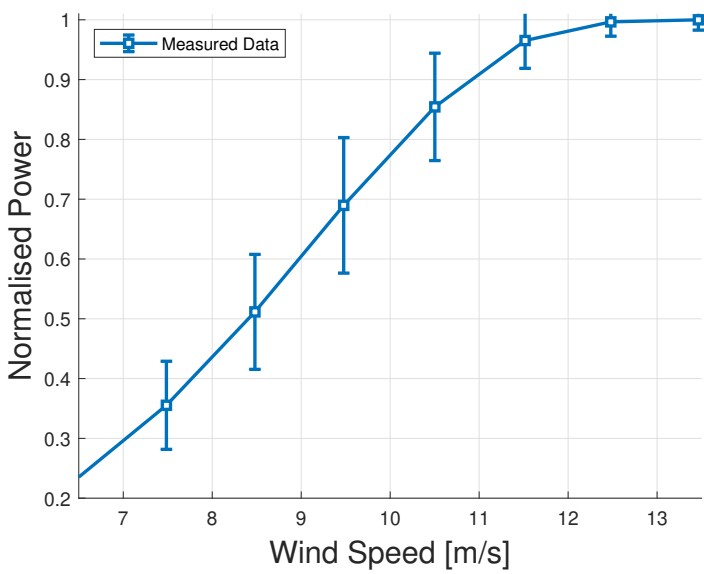

**Figure 7.** Power as a function of wind speed (filtered dataset, 10-minute averaged - Measured). Error bars represent one standard deviation from the mean.

### 3.2.1 Seasonal trend decomposition

The seasonal trend decomposition analysis of the TPI signal, performed in this study, builds upon the methodologies and findings presented in Malik and Bak (2024b). While the fundamental approach to decomposing turbine performance data into trend, seasonal and residual components remains consistent, the current investigation introduces a nuanced examination tailored to the unique operational characteristics and sensor configurations of the turbine under investigation. The focus of this analysis, is the extrapolation of the previously introduced methodology, paired with a turbine and controller-specific sensor pair, i.e., power as a function of wind speed, based on simulation-based results (see Section 3.1).

Figure 8 illustrates the trend decomposition of one of the sixteen turbines under investigation. This figure illustrates the decomposition of a single turbine's performance, highlighting the long-term performance improvement or decline, the recurrent seasonal patterns and the short-term deviations from expected performance trends. Here an increased trend reflects improved turbine performance and the opposite for a reduction in trend trajectory. These changes may be caused by operational and maintenance events, blade repair, erosion as well as various other causes. The seasonal component illustrates the cyclical performance variations attributable to environmental factors. It is worth noting that the analysis methodology has been applied in scenarios including waked turbines, yielding consistently robust results despite the potential for additional variability in those conditions. Importantly, the TPI signal relies exclusively on data generated by the individual turbine, without incorporating comparisons to neighbouring turbines or meteorological masts.

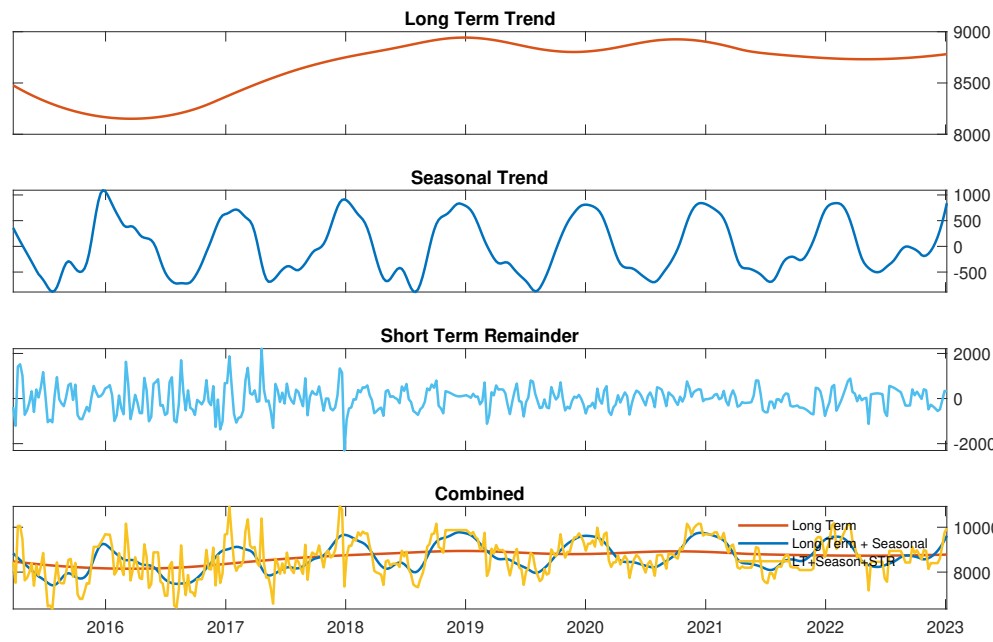

**Figure 8.** Decomposition of a single turbine's performance trends over nine years - Power as a function of wind speed. Vertical scales represent turbine performance integral (TPI) in units of Power·Wind Speed (kW·m/s).

To highlight the pivotal role of sensor pair selection, consider the power-to-wind speed TPI signal. This signal, is a more responsive indicator for detecting performance oscillations, which is empirically substantiated here. Figure 9 elucidates the comparative dynamics of TPI signals extracted using two distinct sensor pairs: power as a function wind speed and generator speed as a function of power. The normalisation process, involving the division of the seasonal trend component by the long-term trend component, provides a dimensionless metric encapsulating temporal performance variations. The power-to-wind speed TPI signal exhibits pronounced cyclicality, reflecting substantial seasonal performance fluctuations, demonstrating its superior sensitivity to performance oscillations. Conversely, the generator speed-to-power TPI signal demonstrates a notably muted cyclical behaviour, largely due to the turbine's generator speed adhering to a pre-encoded operational 'ceiling' - refer to Figure 2. This programmed limit delineates the maximum permissible generator speed relative to power, preventing upward deviations.

### 3.2.2 Seasonal influence

Presented in Figure 10 are the aggregated seasonal trends of the investigated turbines, highlighting variations that may not be evident from the analysis of individual turbines. The overlaid individual results provide empirical validation of the turbine

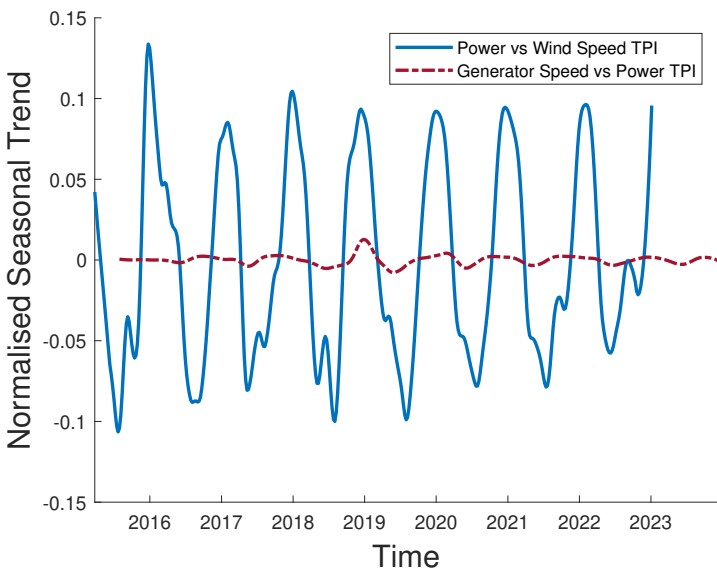

**Figure 9.** Comparison of normalised seasonal trends in turbine performance integral (TPI) over time for two sensor pairs: power as a function of wind Speed and generator RPM as a function of wind speed. The vertical axis represents TPI in W·m/s, with values normalised to highlight relative changes. This comparison demonstrates the superior sensitivity of the Power as a function of wind speed pair.

performance integral (TPI) method, introduced in Malik and Bak (2024b) and demonstrate the efficacy of power curve based
selected sensor. The strong synchronisation evident across the turbine population underscores the suitability of this approach.

While Figure 10 appears dense, its primary purpose is to illustrate the high degree of synchronisation across the entire turbine set rather than to track individual turbine performance. Readers should focus on the overall pattern and synchronicity, which validate the effectiveness of the selected sensor pair and the TPI method.

A notable observation is the tight synchronisation in performance variation signals, particularly during winter peaks and
380 summer troughs, a pattern further delineated in the violin plots (Bechtold (2016) Bechtold et al. (2021)) presented in Figure 11. This synchronisation, exceeding the coherence found in the previous work, Malik and Bak (2024b), could indicate a better-fitting signal pair, despite the power curve incorporating the uncertainty of wind speed. Alternately, this may be attributed to an improvement in the quality of the underlying data with fewer gaps caused by factors such as de-ratings or outage type events. Such improvement in data integrity potentially stems from the weekly data buffering underlying the system, which ensures a
385 more robust outcome - described in Malik and Bak (2024b). However, it is crucial to note that buffering would still introduce 'elasticity' in the signal's representation in cases of missing data, as data bins still require filling.

The results reveal not only the expected seasonal variations but also additional intriguing patterns that warrant further exploration. Specifically, the winter peaks display a characteristic pattern of an initial lower peak, towards the end of the year, followed by a minor trough and then a pronounced peak. Similarly, the summer troughs exhibit a brief peak before descending
further. These patterns appear consistent across most turbines in a given season, but not across all seasons.

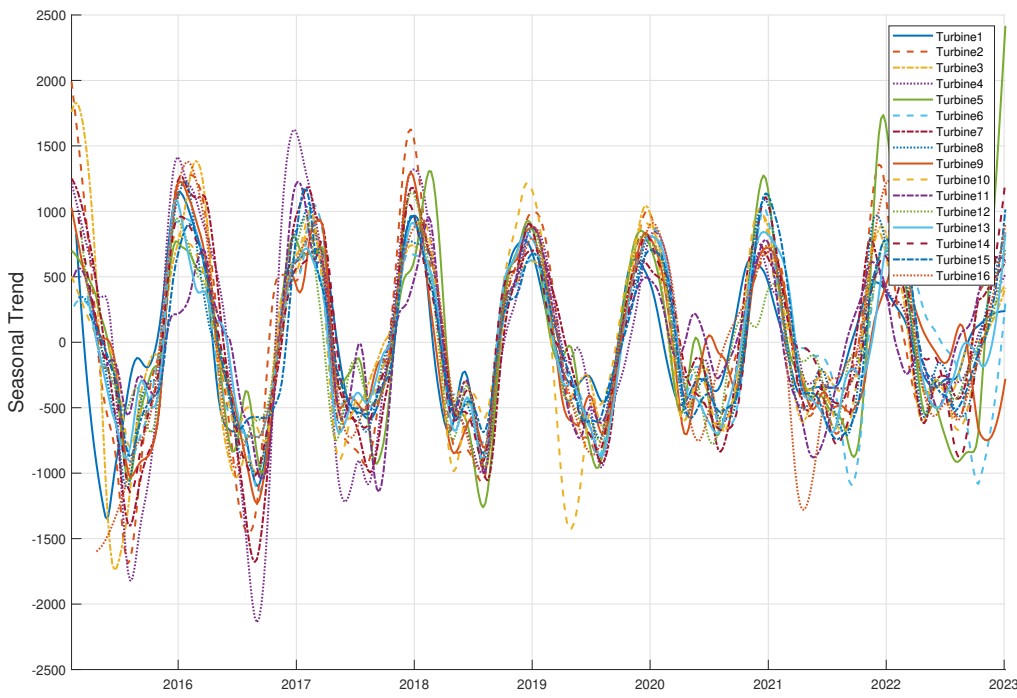

**Figure 10.** Synchronised seasonal performance trends for sixteen wind turbines over time. Each line represents a turbine's turbine performance integral (TPI) in kW·m/s. Higher TPI values indicate better performance. The graph illustrates the high degree of synchronisation in seasonal patterns across the turbine fleet, with clear annual cycles visible.

Since the signal is not normalised for air density variations, unlike the approach in the previous study, the observed variations encompass atmospheric conditions, including temperature as well as wind direction and turbulence. These distinct patterns raise questions about the specific meteorological conditions influencing these variations. Future research could focus on identifying correlations between performance patterns and weather data to gain a deeper understanding of the underlying cumulative factors driving these trends. This distinct seasonal trend in turbine behaviour may also reflect a unique signature of the specific site, varying for identical turbines in different locations, conceptualising the turbine as an instrument measuring local atmospheric characteristics.

Moreover, the characteristic patterns within the seasonal trends warrant further investigation, potentially through an interdisciplinary collaboration with meteorologists. Such collaborations could help identify specific atmospheric phenomena driving these performance variations. Alternatively, these additional 'bumps' or minor peaks in data may be mathematical artefacts intrinsic to MATLAB's implementation of STL via the "trenddecomp" function, employed in this work (The MathWorks, Inc. (2023)). Additionally, understanding these patterns could aid in the calibration of sensor data.

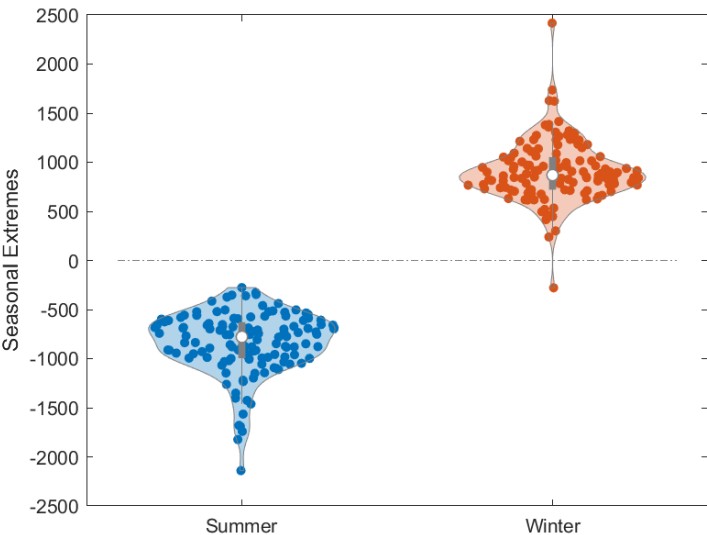

**Figure 11.** Violin plot comparing seasonal performance extremes for sixteen turbines. Summer (left) and winter (right) variability in turbine performance integral (TPI) are shown. Higher TPI values indicate better performance. The plot illustrates the distribution, median, and range of seasonal performance variations across the turbine fleet.

The improved clarity and definition of the seasonal decomposition signal, compared to previous work, offers the potential to derive valuable performance insights. For example, analysing deviations of a single turbine's performance from its historical pattern or from the trends of neighbouring turbines could signal underlying performance issues and pinpoint the need for targeted interventions or maintenance. This emphasises the applicability of seasonal performance analysis as a proactive maintenance tool within wind farms.

### 3.2.3   Long term trend

Figure 12, illustrates the temporal progression of sixteen turbine's long term performance. This visualisation facilitates to understanding the overarching trends and deviations in turbine performance over the extended period, providing insights into the effects of variables such as operations and maintenance, environmental influences and blade erosion on turbine efficiency.

The zeroing of the trend data accentuates relative changes over time, enabling an examination of the performance deviations from a normalised baseline, highlighting those that diverge from the fleet's general performance trajectory.

Turbines 4, 5, 6, 7, 8, 11 and 13 were initially commissioned without LEP, leading to accelerated wear compared to blades with LEP. The subsequent installation of LEP on these turbines at later dates potentially also influences their performance trajectories. Specifics of these LEP installations, including dates, are provided in Section 3.2.4.

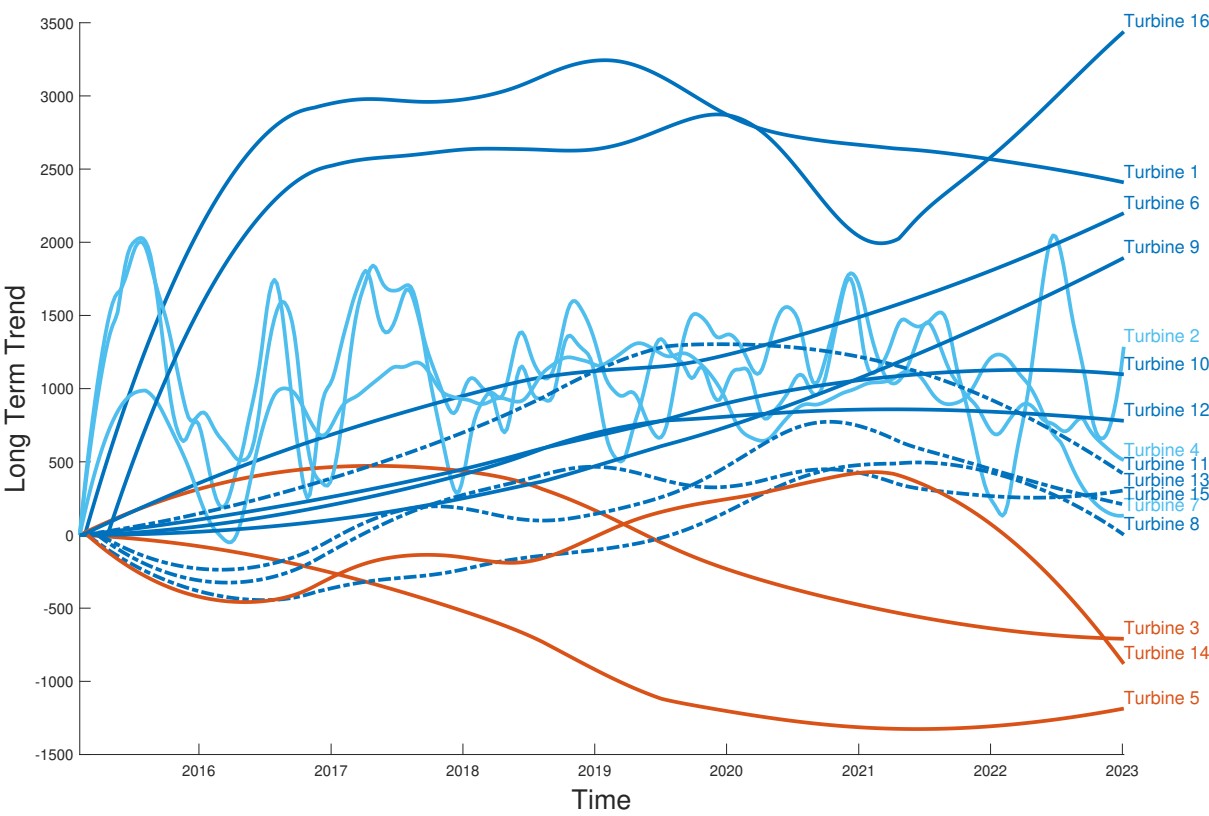

**Figure 12.** Grouped long-term trends in turbine performance: analysis of shared trajectories among sixteen turbines - performance increases with value.

The longitudinal analysis depicted in Figure 12 show a diverse array of performance trajectories across the analysed turbine fleet. Specifically, Group A, Turbines 1, 6, 9 and 16 exhibit an upward trend, potentially indicative of improved performance stemming from successful maintenance interventions or systematic upgrades implemented over the observed period. Con-
versely, Group B Turbines 3, 5 and 14 show a downward trend, suggesting progressive performance decline, possibly due to accumulated wear that maintenance efforts have not fully mitigated. Group C, including Turbines 8, 11, 13 and 15 show a somewhat stable trend.

The variable performance of Turbines 2, 4 and 7, in Group D, characterised by intervals of sharp increases and decreases, aligns with patterns reported in earlier work (Malik and Bak (2024b)). Such fluctuation could result from a combination
of operational dynamics and external environmental factors. Integrating this analysis with meteorological data could help elucidate the underlying causes. Moreover, methodological limitations, such as the application of the STL decomposition might also contribute to these variations. Adjusting the smoothing or other parameters to minimise 'leakage' of seasonal effects into the long-term trend could improve trend fidelity, possibly causing seasonal effects to be visible in the long term trend could enhance trend fidelity and prevent misattributing seasonal effects to climatic variability. A thorough investigation

incorporating the turbines' maintenance history and regional climate conditions, is warranted to clarify their impact on the observed performance dynamics.

Generally, the turbines are noted to improve or maintain performance over the analysed period, with a few exceptions that warrant further investigation. While a detailed comparison with Malik and Bak (2024b) is beyond the scope of this analysis, the identification of similar patterns underscores the value of longitudinal performance assessment. This approach aims to facilitate data-driven decision-making for maintenance and contributes to understanding factors influencing wind turbine performance over time.

### 3.2.4 Influence of erosion and blade operations and maintenance events

Informed by the synchronised seasonal trends that emphasise the importance of turbine-specific sensor selection, this section explores the impact of LEP applications and repairs on a targeted subset of turbines' long-term performance. Figure 13 and subsequent Figures A1 and A2, shown in the Appendix A, illustrate these effects.

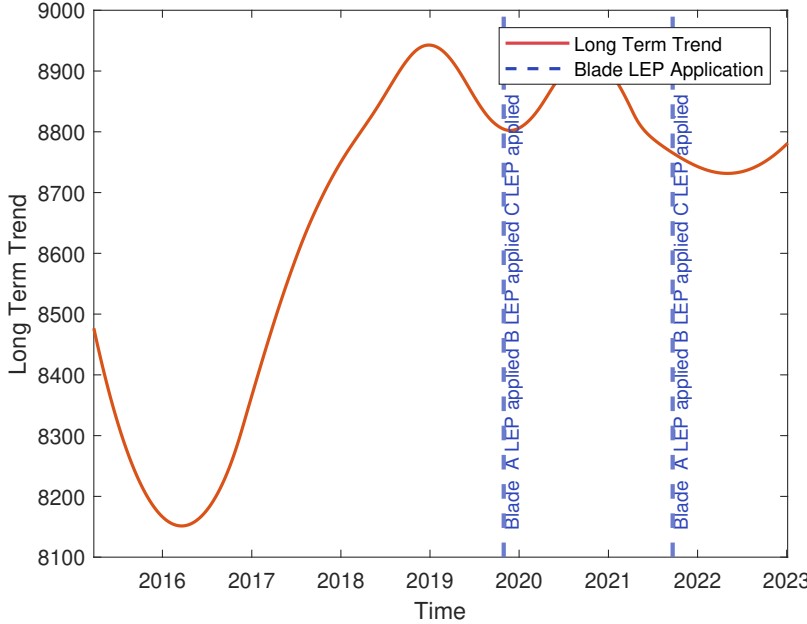

**Figure 13.** Overlay of blade maintenance activities on long-term turbine performance integral (TPI) trends. Performance increases with higher TPI values. Vertical dashed lines indicate blade leading edge protection (LEP) application. The solid line represents the long-term TPI trend.

While blade-related interventions and erosion have the capacity to alter turbine performance, a multitude of other unaccounted-for factors also contribute to deviations. These include weather events, O&M events, component replacements, control system updates, measurement uncertainties and more. A comprehensive effort to document every influencing factor and its impact, is undertaken in Malik and Bak (2024b). However, the extensive data aggregation required and the potential for inconclusive

results, is not replicated here due to the extensive data aggregation required and the potential for inconclusive results stemming from insufficient event data in that work.

This study's further focus is identifying turbine-specific critical sensors, as evidenced by the synchronised seasonal trends. Despite the thorough analysis, erosion detection does not yield definitive conclusions, necessitating the exploration of alternative methods. In the subsequent sections, potential sensors suitable for detecting erosion shall be evaluated.

### 3.3 Refined multibody simulations for detailed sensor evaluation

Motivated by the limited sensors availability in operational studies based on SCADA data, this investigation revisits the multibody simulation environment to examine the response of various sensors to blade roughness.

Figures 14 and 15 exemplify the changes in electrical power, attributable to two distinct degrees of blade roughness, as a function of wind speed and for various turbulence intensities. The impact of erosion becomes markedly perceptible at wind 455 speeds exceeding 9 m/s, with the P40 roughness having a more pronounced effect on the power curve. Moreover, the influence of erosion is more pronounced at lower turbulence intensities, as evidenced by the most significant change in power at $0\ \% \ TI$ compared to $12\ \% \ TI$. An annual mean $TI$ of 6% is considered representative for the offshore site under investigation. This aligns with the anticipated impacts of erosion on aerodynamic efficiency and, consequently, turbine sensor readings.

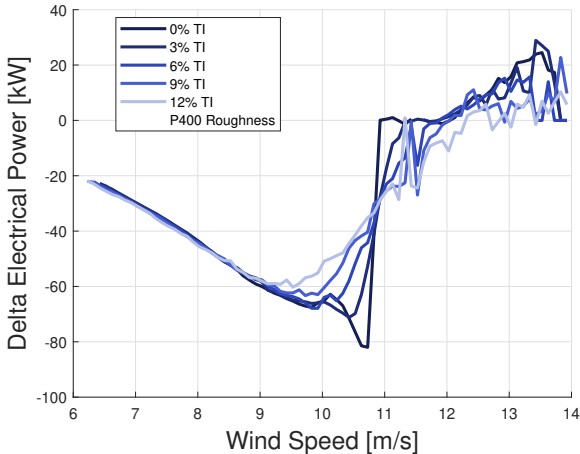

**Figure 14.** Delta electrical power due to P400 leading edge roughness compared to a clean blade, for various turbulence intensities $(TI)$.

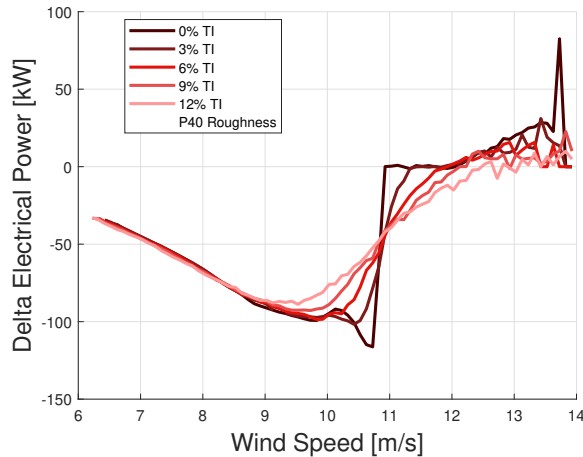

**Figure 15.** Delta electrical power due to P40 leading edge roughness compared to a clean blade, for various turbulence intensities $(TI)$.

To quantify the sensitivity of various sensors to blade erosion, Cohen's $d$ was selected as the metric of choice to provide a 460 standardised and interpretable measure of the effect size of blade erosion (P40 roughness). This metric allows a comparison of the responsiveness of different sensors across varying wind speeds and turbulence intensities, providing insights into which sensors are most effective for detecting blade erosion. In Figure 16, the heat map provides a visual representation of Cohen's

$d$ values, demonstrating the differential sensitivity to erosion across varying wind speed bins for a limited suite of sensors, at a turbulence intensity of 6 %. The results for 0 % and 12 % are provided in Appendix B, Figures B1 and B2, respectively.

Figure 16 presents a comprehensive heat map of Cohen's $d$ values for multiple sensors across different wind speeds at a $TI$ of 6 %. This visualisation is crucial for identifying the most sensitive sensors and the wind speed ranges where erosion effects are most pronounced. To interpret the heat map:, observe the x-axis, which represents different wind speed bins and the y-axis, which lists the various sensors being evaluated. Each cell in the heat map corresponds to the Cohen's $d$ value for a given sensor at a particular wind speed. Warmer colours indicate higher magnitudes of change, suggesting greater sensitivity of that

sensor to blade erosion, while cooler colours indicate lower magnitudes of change. This visual representation allows for quick identification of the most responsive sensors across different operational conditions.

     To focus on magnitude rather than direction of changes in sensor readings due to blade erosion, the absolute values of Cohen's $d$ are taken, extending the range from 0 to 2. This adjustment simplifies the interpretation of results, as it emphasises the extent of change rather than its direction. Moreover, the values within this range are not displayed in the figure; the figure

serves solely as a guide to identify which sensors and wind speed regions warrant further analysis.

     While the absolute values of Cohen's $d$ typically range from 0 to 2, it is important to interpret them in the context of the specific sensor. As an example, the response of electrical power (6 % $TI$ P40) is directly relatable to Figure 15, herein presented in terms of Cohen's $d$.

     Higher absolute values of Cohen's $d$ suggest a greater sensor sensitivity to blade erosion. The colour scale ranges from

0 to 2, with darker colours representing greater change in value. A value of 0 indicates no difference between clean and rough conditions. The heat map colour scale was limited to this range to improve the visualisation of patterns across sensors, highlighting relative differences and making patterns easier to discern. While this obscures the absolute difference in sensor response, a logarithmic scale could compress the range of Cohen's $d$ values, although it would make interpreting the effect's magnitude less intuitive.

Sensors registering the most substantial Cohen's $d$ values across multiple bins warrant particular attention in relation to the research question. The Cohen's $d$ values for torsion at the blade tip were exceptionally higher in magnitude compared to other sensors. Values reaching approximately -13 (6 % $TI$) suggest either a substantial sensitivity of blade tip torsion to blade erosion conditions or potential overestimation of this sensitivity by the model. Further analysis of the blade tip torsion data is needed to determine the primary cause. The underlying torsion data may have extreme values or outliers (for both rough and clean

conditions) that might be skewing the results. It should be consider whether the simulation model might be overemphasising the blade tip torsion response under certain conditions. Additionally, if the standard deviation of the blade tip torsional load is particularly small within conditions, even moderate differences in means can produce a large Cohen's $d$.

     The heat map analysis reveals sensors with marked sensitivity to erosion, specifically blade tip torsion, blade root flap moment, shaft moment and tower moments. These sensors demonstrate particular sensitivity under lower turbulence intensities

- comparing Figures B1 and B2. However, care should be taken in practical application with sensors such as the tower bottom moment, which may not be as reliable in a real-world environment as in simulations. This sensor's distance from the primary cause of the effect, blade erosion, can result in significant noise interference. For instance, fouling on the foundation, which

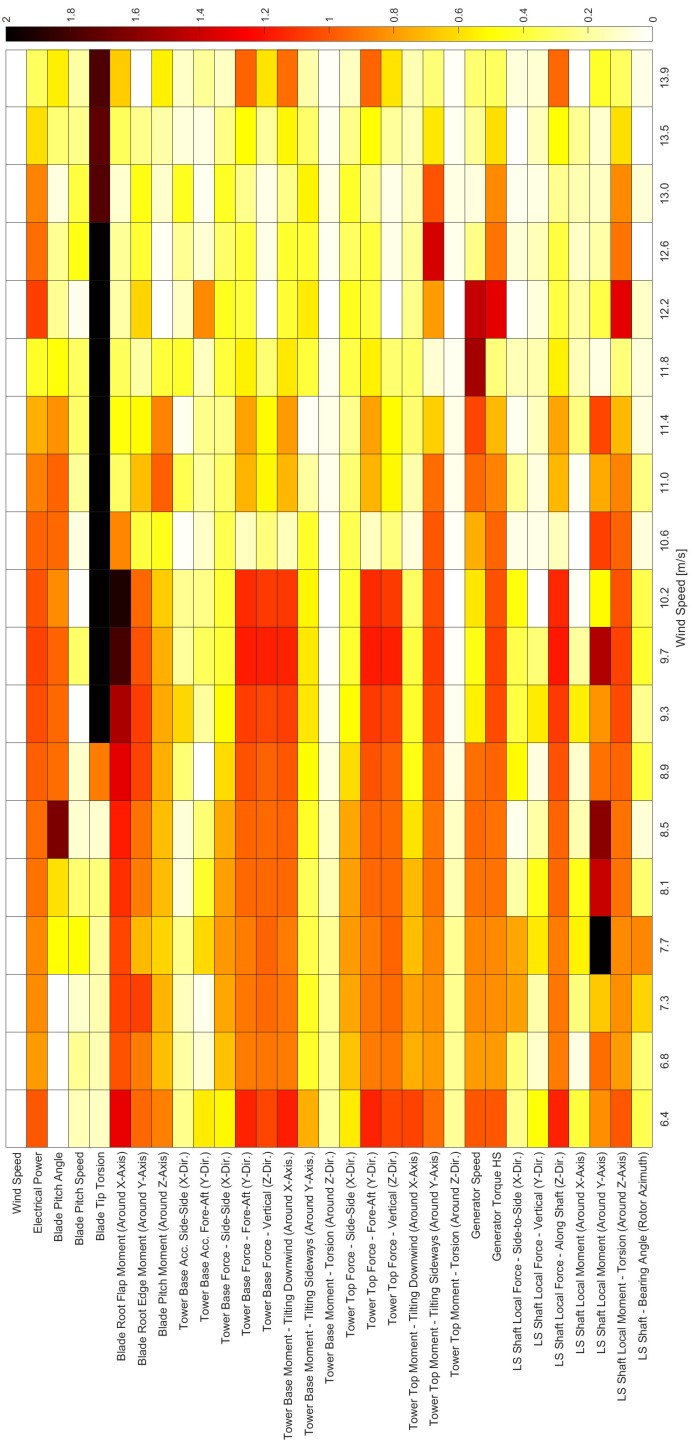

**Figure 16.** Heat map of Cohen's $d$ values showing sensor sensitivity to blade erosion (P40 roughness vs. clean) across wind speeds at 6% $TI$. Warmer colours indicate higher sensitivity. Cohen's $d$ absolute values range 0–2.

may also vary over time similarly to erosion, can confound the readings from such sensors, making it challenging to attribute changes directly to blade erosion.

Although, the heat map analysis reveals several key findings, it is crucial to acknowledge that these results are based on multibody simulations, which may have limitations in representing non-uniform inflow conditions (Boorsma et al. (2024)). Additionally, the aerofoil aerodynamic model may have reduced accuracy at the high Reynolds numbers, as limited validation exists for eroded aerofoil modelling at these conditions. These limitations may affect the accuracy of the sensor sensitivity analysis.

These findings provide insights into the capabilities of various sensors for erosion detection and performance monitoring. They emphasise the potential utility of sensors that may show promise for integration into existing SCADA or condition monitoring systems (CMS). This integration may enable the detection of both blade erosion and performance alterations due to other potential blade aerodynamic profile change related causes.

Furthermore, these findings suggest potential benefits for wind farm owners and operators in discussing sensor inclusion with
turbine manufacturers during contract negotiations. Certain sensors, such as those embedded in the drive train or blade layup, are typically installed during manufacturing and difficult to retrofit later. Access to data from these sensors at a appropriate sampling rates through standard SCADA systems could strengthen fleet monitoring capabilities. Owners and operators may want to consider requesting such access to improve their ability to monitor turbine performance over time.

## 4   Conclusion

This investigation explores advancements in assessing wind turbine performance using blade erosion as a proxy for detrimental performance changes. The work describes the process of utilising a turbine OEM-provided multibody model for effective sensors selection on the same operational offshore wind turbine. Notably, the turbine's wind speed anemometer, previously considered of limited utility, appears to be a crucial sensor for performance monitoring. However, the inherent uncertainties in wind speed measurements must be accounted for when interpreting performance trends, as they could significantly influence
the reliability of data-driven insights.

The study applies the turbine performance integral (TPI) to a multi-megawatt turbine of a different manufacturer than in previous work (Malik and Bak (2024b)), testing the TPI's effectiveness across diverse operational contexts. This suggests the necessity of a controller-informed, turbine-specific approach to sensor selection and highlights the potential benefits of collaboration between turbine manufacturers and operators. Such partnerships are crucial for applying proprietary control
philosophies to deploy the most appropriate sensors.

This research attempted to bridge the gap between simulation and operational reality by empirically examining the efficacy of an identified sensor pair in an operational turbine. Multibody simulations were used in establishing the correct sensors, which were applied in analysing seasonal performance variations. The analysis shows TPI synchronisation across 16 turbines, in the same wind farm, over a nine-year period, revealing overarching seasonal trends and sub-seasonal variations warranting
further exploration.

However, attributing long-term performance changes to blade erosion or LEP interventions remains challenging. The multitude of operational events throughout a turbine's lifetime often obscure direct correlations between performance deviations and specific interventions. This difficulty aligns with findings from Malik and Bak (2024b), which demonstrated the inherent challenges in drawing correlations between various events in a turbine's lifetime and its performance.

To address these challenges, the investigation returned to the simulation environment. By employing Cohen's $d$ as a normalised metric identified, additional useful sensor signals were identified for the investigated turbine. Blade tip torsion, blade root flap moment, shaft moment and tower moments exhibited heightened sensitivity to blade erosion, particularly under lower turbulence intensity conditions.

While the insights gained from the simulation results could not be directly compared with operational data due to lack of access or the potential non-existence of certain sensors, this area presents opportunities for future iterative validation with results compared against empirical results to further refine the methodology. Such refinement must also consider how uncertainties in measurements impact the derived trends, particularly when the wind speed signal is employed by TPI. This may involve adjusting the simulation parameters, refining the sensor selection criteria, or incorporating additional data processing techniques. The iterative approach aims to ensures that the final set of identified sensors is both theoretically sound and practically relevant. The goal is to converge on a set of sensors that exhibit strong correlations with performance trends in operational data, potentially improving erosion monitoring. It is important to note the limitations of this approach, particularly regarding potential inaccuracies of multibody simulations in non-uniform inflow conditions. The ultimate aim is to identify reliable and practical indicators of blade erosion-related performance changes that could be implemented in real-world turbine monitoring systems.

The study indicates the pressing need for widely-available turbine-specific simulation models that accurately reflect operation under real-world conditions. Such models could be useful for fine-tuning sensor selection and deepening the understanding of turbine performance nuances. This analysis of simulated sensor effectiveness in detecting performance reductions due to blade erosion has several potential implications for wind turbine operation and maintenance:

– Tailored sensor selection: Operators may be able to improve performance monitoring accuracy by focusing on specific sensors with high sensitivity to blade erosion, as determined through turbine-specific models.

– Sensor sensitivity: This research suggests that certain sensors are particularly sensitive to surface roughness caused by erosion. Their high Cohen's $d$ values indicate their potential for early detection of performance degradation. The heightened sensitivity at lower turbulence intensities suggests the value of filtering datasets for calmer wind conditions to improve the likelihood of detection.

– Potential for early detection, optimised maintenance and enhanced efficiency: Integrating highly sensitive sensors into existing SCADA or CMS systems could enable proactive maintenance scheduling, potentially minimising energy losses and preventing severe damage. The work identifies potential sensors that may provide the most reliable indicators of erosion-related performance changes, supporting data-driven decision-making for improved operational efficiency and asset longevity.

Increased collaboration between academics, turbine OEMs and operators appears to be important, promoting data-driven strategies to improve performance monitoring accuracy. This collaboration may facilitate the practical application of research findings and provide insights for future studies aimed at advancing the sustainability and efficiency of wind energy production.

**Appendix A: Influence of erosion and operations and maintenance events for all sixteen turbines**

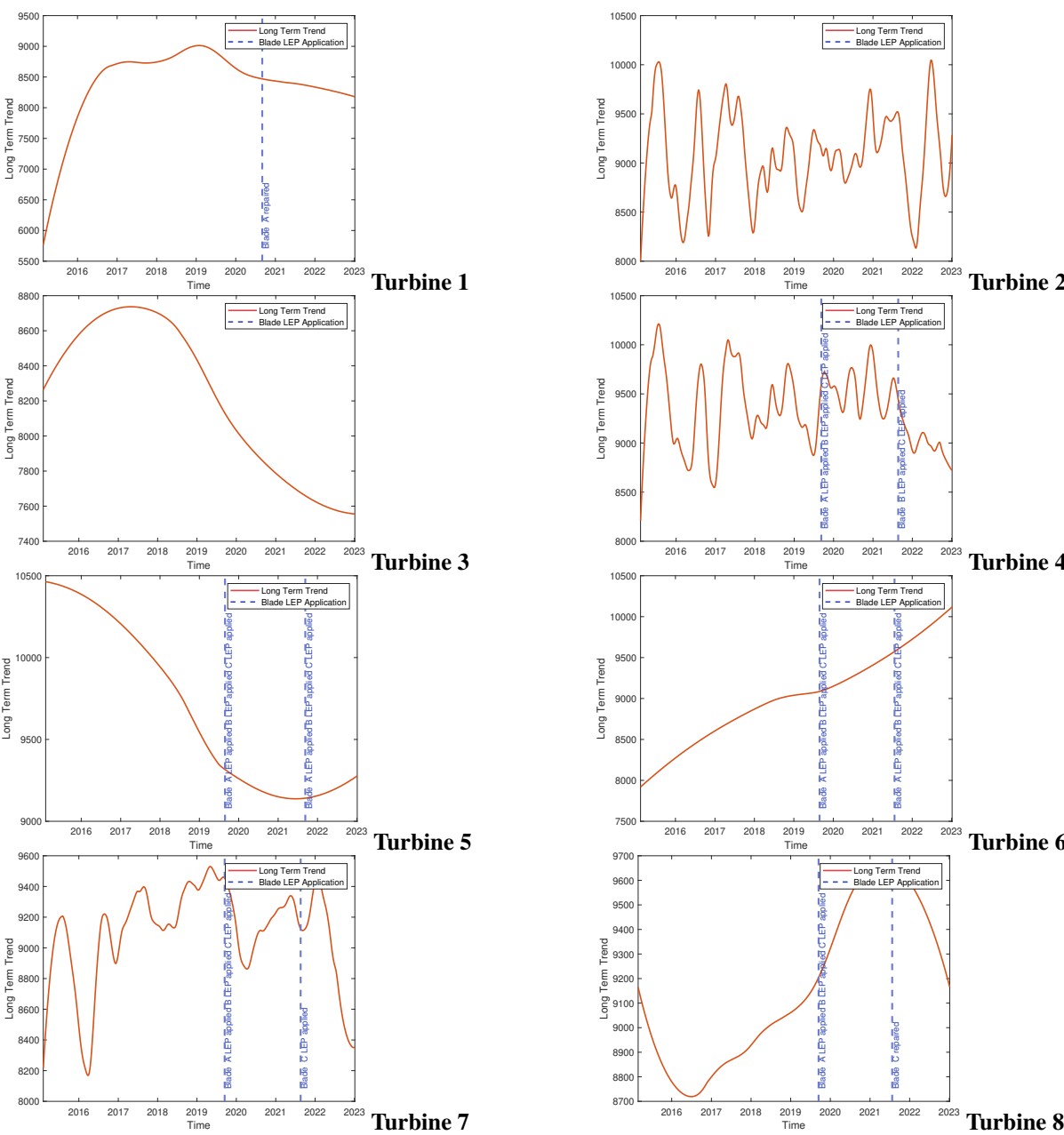

**Figure A1.** Overlay of blade maintenance activities on long-term turbine performance integral (TPI) trends. Performance increases with higher TPI values. Vertical dashed lines indicate: blade leading edge protection (LEP) application. The solid line represents the long-term TPI trend.

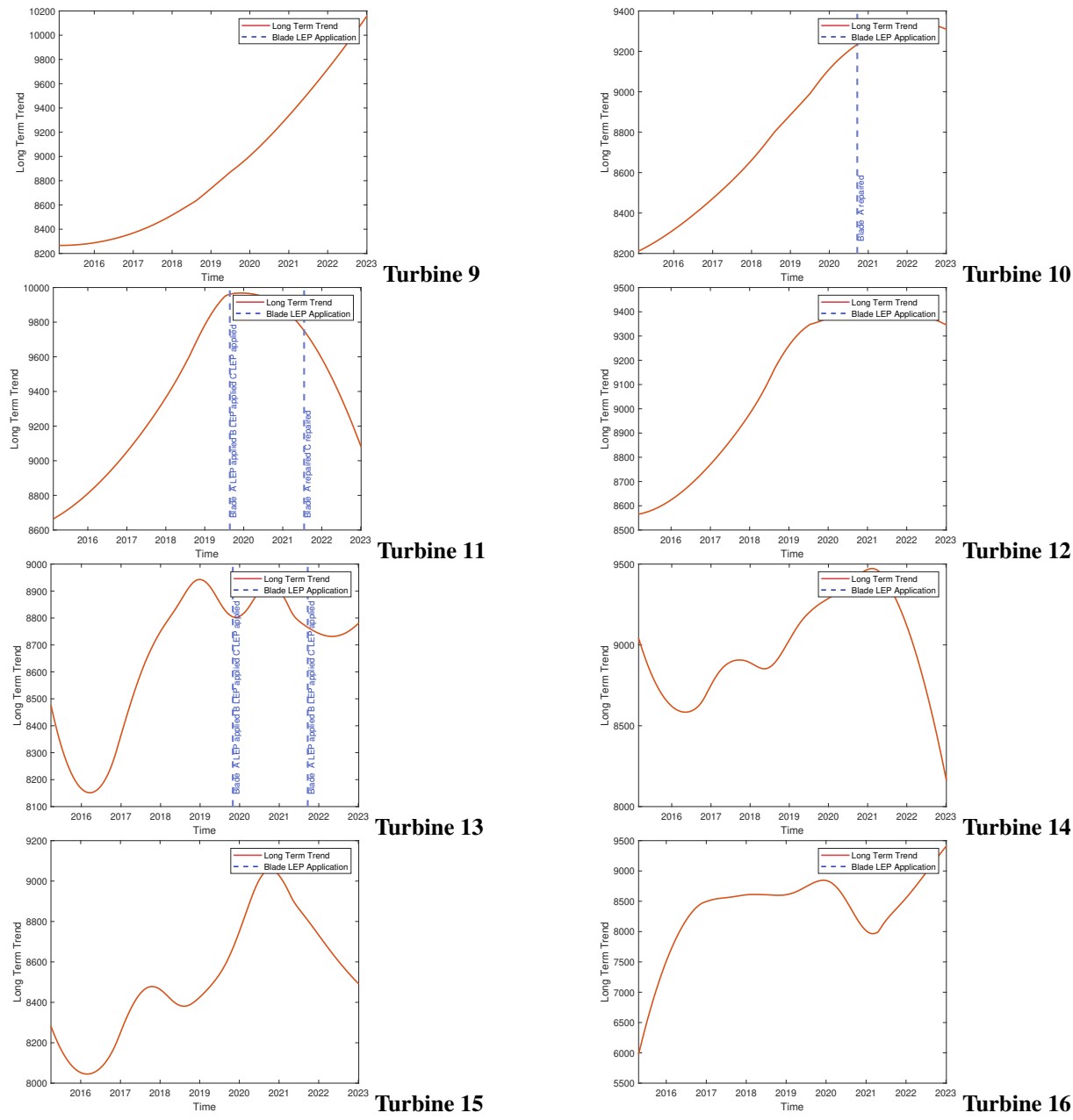

**Figure A2.** Overlay of blade maintenance activities on long-term turbine performance integral (TPI) trends. Performance increases with higher TPI values. Vertical dashed lines indicate: blade leading edge protection (LEP) application. The solid line represents the long-term TPI trend.

**Appendix B: Cohen's $d$ as a function of wind speed tough (P40) - clean, for multiple sensors, at various turbulence intensities**

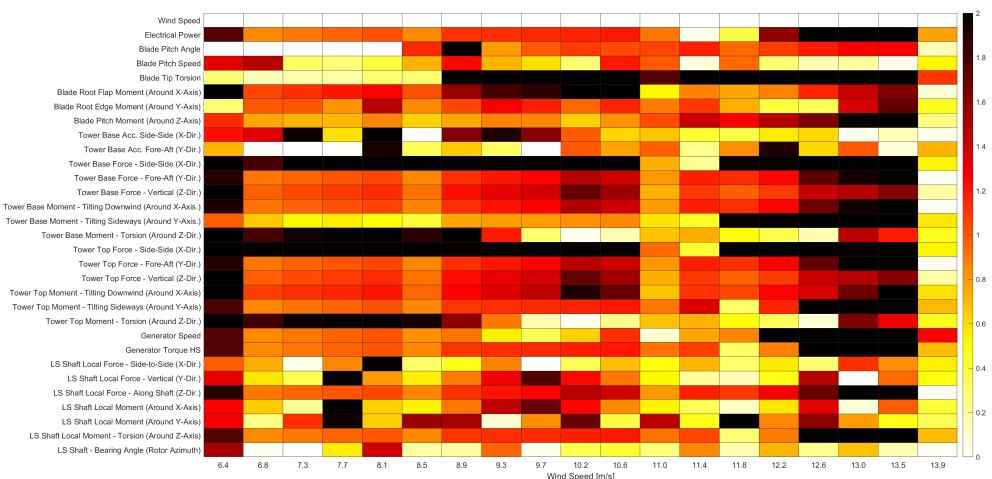

**Figure B1.** Cohen's $d$ as a function of wind speed rough (P40) - clean, for multiple sensors, at $0\%$ $TI$.

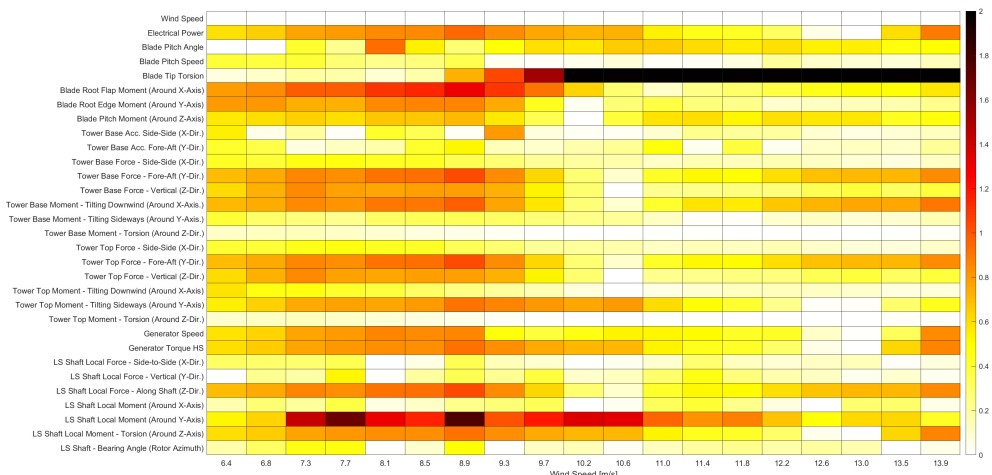

**Figure B2.** Cohen's $d$ as a function of wind speed rough (P40) - clean, for multiple sensors, at $12\%$ $TI$.

*Author contributions.* THM was the primary researcher, responsible for the conception of the study, all experimental work, data collection, analysis and writing the paper. CB, as the PhD supervisor, provided oversight, theoretical support and guidance in refining the research methodology and paper.

*Competing interests.* The author Tahir H. Malik's PhD was funded by Vattenfall, where he is also employed.

*Acknowledgements.* We gratefully acknowledge the support of Vattenfall, particularly for financing this study and providing access to vital wind turbine resources. We also thank the AI language model, OpenAI (2024), for refining a previous version of the manuscript.

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
