# Peer review of "Full scale wind turbine performance assessment: a customised, sensor-augmented aeroelastic modelling approach"

_Wind Energy Science, 2024_

## Referee Comment (RC2)

This article showcases a practical application of wind turbine blade erosion detection using a combination of aero-elastic simulations and real-world SCADA data. To me the article definitely has the potential to meet the journal's scientific standards but in its current form the balance between application and scientific content leans too much towards the application and in some sections the language even tends to be a bit promotional

Hence a more complete description of the methodology is essential for a proper evaluation of the scientific merit of the article.

I agree to the comments of the first reviewer but in addition the following recommendations should be considered

- The current study can hardly be understood without knowing the previous reference presented in Malik and Bak (2024a). A brief summary of their key results would be beneficial for readers unfamiliar with that reference.

- While I understand the need for confidentiality regarding specific turbine and site details, some generic information, such as the turbine size class (e.g. multi-megawatt offshore) is essential to put the findings in context and understand the general validity of the approach. Perhaps the interpretation of the results differs significantly for a study involving kilowatt-scale turbines compared to multi-megawatt offshore installations? I now read between the lines in section 2.1.2 that the turbines are off-shore and in the conclusions I read that they are Multi MW. Please disclose this information upfront. This is also needed to interpret the absolute numbers in section 2.1.1. (outer 9 m of the blade, roughness numbers etc).
Related to this: What is a typical Reynolds number? To me the aerodynamics of erosion depends heavily on the Reynolds number.

- The study relies on results from HAWC2 simulations. While HAWC is a well-validated aeroelastic modeling tool, a scientific sound approach requires an assessment of validity and possible limitations of the modelling approach for the current situation. Specifically, it would be helpful to understand whether any known inaccuracies identified in e.g. Boorsma_2024_J._Phys._Conf._Ser._2767_022006.pdf (dtu.dk) might impact the findings. The same holds for the accuracy of the airfoil aerodynamic model used, particularly bearing in mind the potentially high Reynolds numbers (above 10 million) for which limited validation of modelling approaches for eroded airfoil are carried out

- Justify (or reframe from) unfounded statements to avoid a tendency of subjectivity. For example, line 53 states "this study leverages the turbines' own wind speed anemometers, which are often overlooked due to uncertainties". I think many people do see the value of turbine anemometers for various applications so please justify this statement or add a more objective phrasing e.g: "The importance of turbine anemometers, to support erosion detection has been demonstrated" or something like that.

- Check whether all concepts been introduced and put in context, e.g. what is Shell A and Shell B at line 116. Also the partial and complete coverage of 4.5 m is not placed in context.

- The text should be checked on clarity, completeness and readability. For instance, the vertical axis of Figure 9 currently lacks a label specifying the quantity. Additionally, the numerous abbreviations throughout the text are confusing. It would be helpful to include a list of all abbreviations with their definitions.

By addressing these points and the points of the other reviewer, the authors will deliver a strong practical contribution to erosion detection where at the same time the journal's scientific standards are met.

---

## Author Response (AR1)

Dear Peer Reviewer,

Thank you for your valuable and insightful comments on our manuscript. Your feedback has been instrumental in improving the quality and clarity of the paper.

Attached, you shall find a detailed response to each of your comments, along with an updated version of the manuscript reflecting these improvements.

Thank you once again for your thorough review and constructive suggestions.

Best regards,

Tahir Malik

**RC1: 'Comment on wes-2024-49', Anonymous Referee #1, 06 Jun 2024**

The paper presents a comprehensive study on wind turbine performance assessment using a sensor-augmented aeroelastic modeling approach, specifically focusing on blade erosion. It combines HAWC2 aeroelastic simulations with real-world operational data to identify effective sensors for monitoring blade erosion effects. The study emphasizes the importance of customized sensor selection tailored to specific turbine models and control systems. With some enhancements in detail, this paper could set a strong precedent for future research in wind turbine monitoring systems.

There are several points that require improvement for the paper to meet publication standards. Below are detailed comments and suggestions for enhancing the manuscript.

1. The methodology section contains extensive descriptions of previous studies by other researchers and the authors' past work, which dilutes the focus on the current study's innovations. I suggest that the authors summarize previous studies more concisely to provide a brief background and context. Focus on highlighting the unique contributions and innovations of the current study. Emphasize what distinguishes this work from existing literature.

> - Introduction updated to include
>     - summaries of previous work
>     - sharper description of motivation of current work
> - Method
>     - Updated to summarise previous studies and highlight current contributions
>     - Expanded to provide more detailed method to lean less on previous study

2. Some sections of the methodology, particularly the simulation settings and test cases, could be expanded to provide more clarity. For instance, more details on the HAWC2 model parameters and the specific conditions simulated would enhance reproducibility. The treatment of SCADA data and the specific algorithms used for analysis (e.g., STL decomposition) should be described in more detail. Elaborate on the simulation settings, including the exact parameters and conditions used in HAWC2, to ensure the study can be replicated by other researchers.

> - Method: Section 2 is now expanded to provide a more detailed explanation of our methodology for clarity and reproducibility. This includes the treatment of SCADA data and STL decomposition, described in more detail.
> - Concise text added such as HAWC2 simulation settings to clarify distinction and a reference added.
> - The comparison between *simulation and empirical observations* is relevant for Section 2.1 for sensor pair identification **not** Section 2.3 – correction made.

3. One of the main innovations claimed by the authors is the combination of simulation and actual measurement data to identify sensors sensitive to blade corrosion. However, the methodology section, particularly Section 2.3, lacks detailed descriptions of this process. I suggest that the authors provide a clear, detailed explanation of how the combination of simulation and measurement data is implemented. Describe the iterative approach mentioned in Section 2.3 in specific terms, outlining each step of the methodology, the tools used, and the

rationale behind the chosen methods. This will enhance the transparency and reproducibility of your study.

4. Section 2.3 of the methodology appears to be incomplete or missing crucial details. Statements like "The primary objective of this section is the rigorous evaluation of numerous sensors' potential" and "The work, therefore, employs a comprehensive methodology that employs an iterative approach" are vague and lack substantive information. Please confirm the completeness of Section 2.3 and ensure it contains all necessary details. Specifically, provide a step-by-step guide on the rigorous evaluation of sensors, including the criteria for sensor selection, the process of integrating simulation with real-world data, and examples of how this methodology was applied in your study. Clarify any ambiguous statements to improve the reader's understanding.

> RE Section 2.3: Upon careful review, it has come to our attention that certain aspects of the section may have been subject to interpretation. In the interest of maintaining the highest standards of accuracy and transparency, we have taken the opportunity to refine and correct the language to more precisely align with the scope and outcomes of this study. This adjustment serves to enhance clarity and ensure that readers have a well-calibrated understanding of the research parameters and deliverables. Therefore the text has been revised and corrected.

> To enhance clarity, it is important to specify that the phrase "*employs a comprehensive methodology that employs an iterative approach, integrating theoretical simulation and empirical validation to ensure that the findings are anchored in both theoretical rigour and operational relevance*" requires further refinement. Specifically, it should be noted that there is no iteration of the simulated results to compare with empirical results from a turbine due to lack of access or existence of the broader sensor set. The language has been adjusted to align with the actual methodology employed, thereby correctly managing the readers expectations. Additionally, we have included further details on the criteria for sensor selection and provided an expanded explanation to ensure comprehensive understanding. The iteration of the broader sensor set shall be a suggestion for further work.

The section now provides a more detailed explanation of our methodology, including:
A clearer description of the sensor selection criteria, emphasizing relevance to blade aerodynamic performance, availability in existing systems, sensitivity to erosion-induced changes, and signal-to-noise ratio.
An outline of our approach, including the selection of virtual sensors, conducting simulations under various conditions, data processing and analysis using Cohen's *d*.

We have also improved the overall flow and clarity of this section, removing vague statements and providing more details. The revised section now offers a more transparent and reproducible account of our methodology, addressing the concerns raised in your review.

5. The results are presented with limited interpretation and explanation. Several key findings are stated without providing sufficient context or analysis to help the reader understand their significance. The information in some figures in the article is too dense, such as Figure 9 and 15, and the description of the figure information is insufficient before and after some figures, preventing the reader from fully understanding the relevance of the figures and how to draw

relevant conclusions. The text arrangement of some figures in this paper needs to be optimized, such as Figure 12, A1, and A2.

➢ RE Figure 9: Thank you for your comment on Figure 9's density. While the figure is complex, its primary purpose is to illustrate the high degree of synchronization across the entire turbine fleet, not to track individual turbine performance. We've added explanatory text to guide readers and text to the caption. This additional context directs readers to focus on the overall trend and synchronicity, which are key findings of our study, validating our methodology across multiple turbines.

➢ RE  Figure 15: We acknowledge that the information presented is dense, which is intentional given the comprehensive nature of the analysis. However, we recognize the need for improved clarity in its interpretation. To address this:
  • We have added a paragraph explaining how to interpret the heat map, including the use of absolute values and the purpose of the colour scale.
  • We emphasize that the figure serves as a guide to identify sensors and wind speed regions for further analysis, rather than providing exact values.
  • We've included additional context relating the Cohen's d values.

➢ Captions for Figure 12, A1 and A2 improve to clarify and compensate for text arrangement of figures

➢ Captions for various figures have been improved

**RC2: 'Comment on wes-2024-49', Anonymous Referee #2, 22 Jun 2024**

This article showcases a practical application of wind turbine blade erosion detection using a combination of aero-elastic simulations and real-world SCADA data. To me the article definitely has the potential to meet the journal's scientific standards but in its current form the balance between application and scientific content leans too much towards the application and in some sections the language even tends to be a bit promotional

Hence a more complete description of the methodology is essential for a proper evaluation of the scientific merit of the article.

More specifically: I agree to the comments of the first reviewer but in addition the following recommendations should be considered

- The current study can hardly be understood without knowing the previous reference presented in Malik and Bak (2024a). A brief summary of their key results would be beneficial for readers unfamiliar with that reference.

➢ In response we have revised and expanded the Introduction and Methodology Section 2 of the article. We have included concise summaries of key results of previous studies, as indicated by yourself and comments of the first reviewer.

- While I understand the need for confidentiality regarding specific turbine and site details, some generic information, such as the turbine size class (e.g. multi-megawatt offshore) is essential to put the findings in context and understand the general validity of the approach. Perhaps the interpretation of the results differs significantly for a study involving kilowatt-scale turbines compared to multi-megawatt offshore installations? I now read between the lines in section 2.1.2 that the turbines are off-shore and in the conclusions I read that they are Multi MW. Please disclose this information upfront. This is also needed to interpret the absolute numbers in section 2.1.1. (outer 9 m of the blade, roughness numbers etc).

  Related to this: What is a typical Reynolds number? To me the aerodynamics of erosion depends heavily on the Reynolds number.

➢ Thank you for your valuable feedback regarding the need for contextual information about the turbines studied. We have addressed this concern as follows:

  - Added mention of "offshore multi-megawatt wind turbines" to the abstract to provide immediate context.
  - Included "multi-megawatt" between 3 and 4 MW description in the introduction to clarify the scale of turbines investigated as well as average wind speed of approximately 9.49 m/s to address your very fair comments.
  - Replaced absolute numbers with relative percentages (e.g., outer 15% of blade length instead of 9 m) to maintain confidentiality while providing meaningful context.

  Regarding the Reynolds number, we acknowledge its importance in erosion aerodynamics. The text has been updated to reflect this, additionally we have provided the range of the turbines being between 3 and 4 MW and the average wind speed of the size being approximately 9.49 m/s.

It is the authors' intention that the process should be replicated for other turbines such that the results of the method are relevant for the specific turbine under investigation. As such, although the method is intended to be replicated, the results may vary for other turbines. Therefore, while we have restrained from sharing more specific turbine parameters than the ones we have additionally provided, this should be considered a positive aspect of our approach. It encourages readers to reproduce the method for their own turbines in collaboration with OEMs.

Our goal is to promote the inclusion of erosion/performance-sensitive sensors during manufacturing or the retrofit of such sensors, fostering collaboration between academics, operators and OEMs to enhance performance monitoring across diverse turbine models and operating conditions.

- The study relies on results from HAWC2 simulations. While HAWC is a well-validated aeroelastic modeling tool, a scientific sound approach requires an assessment of validity and possible limitations of the modelling approach for the current situation. Specifically, it would be helpful to understand whether any known inaccuracies identified in e.g. Boorsma_2024_J._Phys._Conf._Ser._2767_022006.pdf (dtu.dk) might impact the findings. The same holds for the accuracy of the airfoil aerodynamic model used, particularly bearing in mind the potentially high Reynolds numbers (above 10 million) for which limited validation of modelling approaches for eroded airfoil have been carried out

  - We have made relevant reference to this paper and explained the limitations of the multibody study. Thank you for your comment.

- Justify (or reframe from) unfounded statements to avoid a tendency of subjectivity. For example, line 53 states "this study leverages the turbines' own wind speed anemometers, which are often overlooked due to uncertainties". I think many people do see the value of turbine anemometers for various applications so please justify this statement or add a more objective phrasing e.g: "The importance of turbine anemometers, to support erosion detection has been demonstrated" or something like that.
  - Text updated: *"Furthermore, this study leverages the turbines' nacelle-mounted anemometers, which are otherwise not suitable for power curve documentation due to measurement uncertainties as per IEC 61400-12-1 (\cite{international2017iec}) standard. This standard recommends wind speed measurements at 2.5 rotor diameters upstream."*
  - The text has been fully reviewed to remove aspects of subjective statements and remove promotional aspects. We hope this improves the readability of text, the comment was much appreciated.

- Check whether all concepts been introduced and put in context, e.g. what is Shell A and Shell B at line 116. Also the partial and complete coverage of 4.5 m is not placed in context.

  - The specific type of LEP coverage is not disclosed for Type A LEP while Type B LEP is a shell type. The sentence has been rephrased for clarity.

- The sharing of blade lengths was an oversight on the authors part and has now been updated in relative, percentage, terms.

- The text should be checked on clarity, completeness and readability. For instance, the vertical axis of Figure 9 currently lacks a label specifying the quantity. Additionally, the numerous abbreviations throughout the text are confusing. It would be helpful to include a list of all abbreviations with their definitions.

  ➢ The text has be been reviewed throughout for clarity and readability and removal of subjective statements.
  ➢ Regarding the vertical axis of Figure 9: the caption has been updated.
      o Text has been added: "*the integral, with units of Power·Wind Speed (W·m/s), is indicative of this turbine's performance trajectory and shall be referred to as Turbine Performance Integral (TPI)*"

---

## Author Response (AR2)

Dear Shawn Sheng - Associate Editor (WES),

Thank you for your valuable comments on our manuscript. Your feedback has been instrumental in improving the quality and clarity of the paper.

Attached, you shall find a detailed response to each of your comments, along with an updated version of the manuscript reflecting these improvements.

Thank you once again for your thorough review and constructive suggestions.

Best regards,

Tahir Malik

**Associate editor decision: Publish subject to minor revisions (review by editor), 02 Dec 2024**

Thanks to the authors for addressing the referees' comments and contributing to the journal of Wind Energy Science.

Additional private note (visible to authors and reviewers only):
Please go through the manuscript for possible errors or typos (e.g., line 519, "improve" may be missed before "performance"). Below are a few other specific comments:

line 519, "improve" may be missed before "performance"

➤ Corrected
➤ The paper has been revised throughout to catch typos and text errors and it is hoped that none have slipped.

Abstract: please briefly explain "Cohen's d" for audience not familiar with the metric.

➤ Updated abstract:
*"Refined simulations using various virtual sensors quantified the effect size of sensor reading under different turbulence levels and blade states, employing Cohen's d - a dimensionless metric measuring the standardised difference between two means."*

Introduction: lines 35-39, please elaborate on sensor pair a bit e.g. by giving an example.

➤ Amended text:
*"In contrast to methodologies that generalise sensor pair applications across different original equipment manufacturer (OEM) turbine models, this work emphasises the deliberate selection of a controller-specific sensor pair. For instance, using power as a function of generator speed or power as a function of wind speed indiscriminately across turbines can overlook critical differences in turbine dynamics and control strategies. This strategy underscores the importance of finding the most suitable sensor pairings for each turbine and associated controller philosophy."*

Methodology: line 131, please explain how 0.1 m/s increments were chosen.

➤ Amended text:
*"In contrast to the previous work, where simulations were run at 1 m/s increments, the current study employs a higher fidelity approach. To focus on the turbine's power ramp-up phase (where erosion effects are most likely to manifest) and to ensure that the binning and averaging process of the data did not obscure subtle dynamics, individual cases were run in 0.1 m/s increments between 6.5 and 14 m/s. This increment achieves a balance between fine-scale accuracy and computational efficiency."*